# Advances in the Understanding of Postharvest Physiological Changes and the Storage and Preservation of Pitaya

**DOI:** 10.3390/foods13091307

**Published:** 2024-04-24

**Authors:** Xiaogang Wang, Jianye Chen, Donglan Luo, Liangjie Ba

**Affiliations:** 1College of Food Science and Engineering, Guiyang University, Guiyang 550005, China; wxgangw@163.com; 2College of Horticultural Science, South China Agricultural University, Guangzhou 510642, China; chenjianye@scau.edu.cn; 3School of Biological and Environmental Engineering, Guiyang University, Guiyang 550005, China; luodonglan1991@163.com

**Keywords:** pitaya, quality change, preservation technology, resistance induction, quality control

## Abstract

Highly prized for its unique taste and appearance, pitaya is a tasty, low-calorie fruit. It has a high-water content, a high metabolism, and a high susceptibility to pathogens, resulting in an irreversible process of tissue degeneration or quality degradation and eventual loss of commercial value, leading to economic loss. High quality fruits are a key guarantee for the healthy development of economic advantages. However, the understanding of postharvest conservation technology and the regulation of maturation, and senescence of pitaya are lacking. To better understand the means of postharvest storage of pitaya, extend the shelf life of pitaya fruit and prospect the postharvest storage technology, this paper analyzes and compares the postharvest quality changes of pitaya fruit, preservation technology, and senescence regulation mechanisms. This study provides research directions for the development of postharvest storage and preservation technology.

## 1. Introduction

Pitaya (*Hylocereus*) is a perennial climbing succulent native to the tropical and subtropical forests of Central America and Mexico and is widely grown and commercialized in at least 20 countries due to its drought resistance, high economic potential, and consumer appeal [1,2]. In addition, pitaya is not only low in calories and delicious, but it also plays an important role in medicinal and ornamental applications and has a wide market [3]. Meanwhile, pitaya is not only rich in nutrients (such as carotenoids, vitamins, phenolics, flavonoids, fiber, and terpenoids) [4,5] but also has the role of being an antioxidative, free radical scavenger in the body, with anti-tumor properties, and reducing cardiovascular diseases in humans, making it a favorite for many people [6,7,8]. Due to the short harvest period and concentrated origin of pitaya, it is often harvested in remote mountainous rural areas with limited transportation. This can lead to mechanical injury or disease during postharvest transportation, resulting in water loss and softening over time. Additionally, the fruit’s respiratory activity accelerates nutrient depletion, causing a decline in quality and a serious impact on its edibility. Moreover, the pitaya fruit itself can be affected by pests and diseases, thus the pitaya fruit storage process often leads to softening, browning, decay and ripeness, and other problems [8,9,10].

Physical, chemical, and biological techniques have been used to maintain the quality of pitaya in response to the postharvest deterioration of quality and the problems of pests and diseases. At present, physical, chemical, and biological technologies are emerging and being widely used in postharvesting, packing, storing, transporting, and marketing of fruits and vegetables. Freshness preservation technology delays postharvest ripening and senescence, maintains fruit quality, and prolongs shelf life by reducing the postharvest respiration rate of fruits and inhibiting microbial reproduction [10]. In recent years, the nutritional value and gene expression of pitaya have been more influenced by basic research on the molecular regulation related to postharvest ripening and the senescence of pitaya fruit and phytohormone metabolism. The analysis of gene expression is an important tool for the elucidation of the complex regulatory network of genetic, signaling, and metabolic pathways [11].

The aim of this review is to clarify the basic situation of pitaya, postharvest quality alterations, postharvest pathogens and diseases, senescence regulation and resistance-inducing technological aspects to better understand the means of postharvest storage of pitaya and its shelf-life extension and the prospects of postharvest fresh preservation technologies.

## 2. About Pitaya

Pitaya is a tropical fruit originating from Latin America [12]. Pitaya can be classified into red skin and red flesh, red skin and white flesh, and yellow skin and white flesh based on the color of the flesh and skin [13]. The cultivation and consumption of pitaya has increased dramatically in recent years due to its unique appearance and rich nutritional content [14]. Pitaya is reportedly grown mainly in the Americas, southern Mexico, Costa Rica, El Salvador, Venezuela, Colombia, Ecuador, Panama, Brazil, and Uruguay [15,16,17,18]. Pitaya grown in each region has different characteristics. The current basic situation of pitaya is shown in Table 1.

## 3. Changes in Pitaya Fruit Quality after Harvesting

Deterioration of pitaya quality after harvest has a serious impact on its market sales, hindering the healthy development of this industry. It is mainly concentrated on respiration, transpiration, scaling, cold injury, fruit softening, pulp browning, nutrient and taste losses, and others (Figure 1).

### 3.1. Respiration

Respiration is a physiological process that takes place after the fruit is harvested and has an impact on its storage quality. Respiration in pitaya is increased after harvest, which is an accelerator of fruit senescence [10]. The respiration rate of the fruit decreases with decreasing temperature under low temperature conditions, and Araujo C.S. et al. [22] showed that low temperature can slow down the respiration rate of pitaya. The respiration rate of pitaya decreases with decreasing oxygen concentration. When the oxygen concentration is lower than 10%, the fruit starts anaerobic respiration, which is unfavorable for storing pitaya. When the CO_2_ concentration exceeds 5%, the respiration of pitaya fruit is inhibited [23]. However, excessive CO_2_ concentration in the environment may cause CO_2_ toxicity, which will result in the decay of pitaya fruit. The pitaya respiration rate is minimized by 2 kPa O_2_ + 5 kPa CO_2_ at 6 °C controlled atmosphere (CA) [24]. In addition to temperature and gases, mechanical injury accelerates the gas exchange between the interior of the fruit and the outside world, causing an increase in respiratory intensity. After mechanical injury, cell membranes are easily ruptured, hydrolytic enzymes are released, and wound-induced signaling starts at the injury site and gradually spreads to non-mechanically injured tissues, accelerating synthesis of the relevant respiratory pathway enzymes, which promotes pitaya respiration [25].

### 3.2. Transpiration

Transpiration of pitaya after harvest is primarily the process by which water evaporates from the pores of the pericarp. The main factors influencing the transpiration rate of pitaya are light, temperature, and humidity. Due to inappropriate storage temperature and humidity, the transpiration rate of pitaya is accelerated, the water content is constantly reduced, and excessive water loss causes shrinkage of tissues and organs, deterioration of quality, loss of nutrients, and also seriously affects physiological metabolism [26]. Vegetable oil (Volatile Organic Compounds, VOC) and carnauba coatings (CC) form a microporous film on the pericarp of pitaya to effectively regulate water evaporation, delay weight loss, and maintain the postharvest quality of pitaya [27].

### 3.3. Scale Wilt

The scale is a specific tissue structure of pitaya that effectively delays mechanical damage to the pericarp during storage and transportation, thus protecting the integrity of the pericarp. The scales contain red pigment, anthocyanins, and chlorophyll, and the fruits are green at maturity [28]. However, chlorophyll is degraded during senescence and the fruit rapidly senesces during storage, causing scale yellowing and wilting [29]. Scale wilting indicates the onset of quality deterioration in pitaya, with scale yellowing and wilting occurring within 3 d at ambient temperature and scale base rot occurring within 7 d, followed by scale rot spreading to the pericarp and fruit [30]. A controlled atmosphere (CA) of 2 kPa O_2_ + 5 kPa CO_2_ at 6 °C significantly delayed scale yellowing, rot, and chlorophyll degradation and maintained fruit firmness and soluble solids, according to Ho P et al. [24]. Scale yellowing was reported in pitaya harvested at eight ripening stages and stored at room temperature for 5 days; the reduction in chlorophyll content in scales was negatively correlated with a sharp increase in chlorophyllase, chlorophyll degradation, peroxidase (POD), and demagnesium chlorophyllase activities [31].

### 3.4. Cold Injury

Pitaya’s cold resistance is poor due to its growth environment. Long-term storage at low temperatures can cause brown spots on the peel and browning of the flesh [32]. Low temperatures accelerate cell membrane metabolism disorder, induce an imbalance of reactive oxygen species metabolism, and accelerate fruit senescence [33]. Although cold injury may occur in pitaya stored at temperatures lower than 5 °C, cold stimulation treatment prior to storage can effectively maintain the metabolic balance, inhibit cold injury, and prolong the storage period [34]. Postharvest preservation techniques such as heat stress, phytohormones, and climate-controlled storage can also delay the degree of cold injury to the fruit [35]. In breeding new varieties of pitaya, genetic engineering techniques can be used to manipulate cold-resistance-related genes and improve the fruit’s cold tolerance at the genetic level. Wang et al. [36] and other studies have confirmed that increasing the expression of the phenylalanine lyase (PAL) gene effectively improves the fruit’s cold tolerance.

### 3.5. Softening of Fruit

After harvest, the fruit adapts to environmental changes and strengthens respiratory metabolism. As a result, scales may wilt and soften, and fruit hardness is significantly reduced, accelerating fruit ripening and quality deterioration. During low-temperature storage at 5 °C, pitaya showed slight cold damage symptoms in the pericarp tissue near the skin, followed by gradual fruit softening [37]. However, preheating the fruit at 25 °C for 24 h before storing it at 2 °C for 14 days delayed the softening process [38]. In addition, Yanmei Xu et al. [39] discovered that treating the fruit with fennel aldehyde helped to delay the decline in hardness and softening and improved the antioxidant capacity of pitaya. Furthermore, pitaya is vulnerable to mechanical damage during transportation, which can lead to infection by pathogenic microorganisms. This may cause anthracnose, brown rot, stem rot, or black rot and aggravate the softening and rotting of the fruit [40,41,42,43,44]. As a result, the edible and economic value of pitaya is reduced, which can impact consumer purchasing behavior.

### 3.6. Browning of Fruit Pulp

Enzymes present in the fruit catalyze the production of phenols, which in turn generate brown complexes through non-enzymatic reactions, leading to the browning of the pulp [45,46]. However, pitaya plantations are susceptible to mechanical injury during transportation after remote harvesting, which promotes enzymatic browning of tissues and gradual darkening of pulp color [47]. Under low temperature conditions, there was a significant inhibition of both polyphenol oxidase activity and total phenolic content [48]. To delay fruit browning, methyl salicylate can be used to inhibit polyphenol oxidase activity [49]. Currently, most of the research on the browning of pitaya fruit pulp centers around phenolic content, changes in polyphenol oxidase activity, and preservation techniques. However, there is a lack of studies on the molecular-level metabolic response to browning. In the future, gene editing technology can be used to inhibit or overexpress key regulatory genes of the browning metabolism of pitaya fruit flesh, solving the browning problem at the molecular level.

### 3.7. Nutritional Loss

Pitaya fruit is a sweet-tasting fruit that is rich in nutrients, including soluble sugars, proteins, betaine, vitamin C, acids, ketones, phenolics, and a variety of trace elements and minerals [2,5,6,7,8]. Thus, total soluble solids are reduced, sugar/acid ratio is reduced, and vitamins and phenolic compounds are reduced, which causes nutritional loss [50]. During postharvest storage, the content of soluble sugars, total phenols, and flavonoids significantly decreased in various pitaya varieties [51]. Additionally, the content of betaine decreased with senescence, but bio-preservatives prevented their degradation and maintained the fruit’s nutritional quality throughout storage [52]. Low temperatures, heat treatment, phytohormones, and biochemical preservatives effectively delay decomposition of vitamin C, acid, sugar, phenolics, and others in fruits [53,54]. However, further in-depth studies are needed to address the regulation of synthesis and catabolism of pitaya fruit nutrients at the molecular level during postharvest storage, which will help solve the problem of postharvest nutrient depletion in pitaya fruit at the molecular level.

### 3.8. Flavor Deterioration

Fruit’s unique flavor is formed by the interaction of nonvolatile and volatile flavors. Flavor varies significantly among pitaya fruit varieties and at different stages of ripening [55]. Volatile compounds are synthesized during the growth and development period of pitaya fruit to give the fruit its characteristic flavor. Flavor is also one of the main characteristics that determine whether it is a high-quality fruit [56]. Pitaya is a perishable fruit that can soften quickly when stored above 20 °C, which can cause a decrease in the sugar–acid ratio and a change in flavor [57,58]. Storage of pitaya at 10 °C followed by transfer to 20 °C significantly decreases the concentration of soluble sugar and acidity during the storage process, resulting in a deterioration of the overall flavor and texture [59]. Currently, 34 volatile aroma compounds have been identified in pitaya, including aldehydes, hydrocarbons, alcohols, ketones, esters, and furans, among which aldehydes are the most abundant volatile compounds in pitaya, with hexanal accounting for 92% of the total compounds in this group [60].

## 4. Postharvest Fungi and Diseases of Pitaya

Fungal diseases are a major cause of postharvest damage to pitaya fruit, resulting in decay and a reduced shelf life [21]. The most common types of postharvest diseases that affect pitaya fruit are anthracnose, black rot, and ulcer [61,62,63]. The purpose of this paper is to summarize the types of postharvest diseases affecting pitaya, their symptoms, the pathogens responsible for them, and the techniques used to prevent and control them (Table 2).

### 4.1. Disease and Disease Characteristics

Anthracnose is a prevalent postharvest disease of pitaya, with an incidence of around 50%. It is caused by *Colletotrichum siamense*, *Colletotrichum tropicale*, and *Colletotrichum truncatum*. Symptoms of pitaya anthracnose are characterized by small, light brown spots on the fruit surface that develop into depressions with concentric black spine rings. These spots gradually expand into reddish-brown spots that depress and spread throughout the fruit until it softens and rots [76,77,78].

*Alternaria alternata* causes black rot, a significant postharvest disease of pitaya fruit [79]. The fungus infests the surface and top of ripe fruits, causing yellowing and soft rotting. A large amount of black mold gradually grows, and eventually, the fruits become water-logged, and the surface sinks and rots [79,80,81].

*Neoscytalidium hylocereum* can infect pitaya stems and fruits during growth and development, resulting in pitaya ulcer disease. This disease is characterized by small, rounded, sunken, orange ulcers that are unevenly distributed on the fruit’s surface [63].

### 4.2. Techniques to Prevent and Control Disease

Pitaya is susceptible to external pathogenic microorganisms that can cause complex diseases, such as anthracnose, black rot, and ulcers, resulting in wilting and rotting of the fruit. Studies have shown that *Bacillus subtilis* can inhibit the growth and spore germination of the pathogens that cause anthracnose [64,70]. Immersing pitaya in a 0.1 mM sodium nitroprusside solution for 8 min was found to inhibit the increase in diameter of pitaya fruit spots and reduce the incidence of disease [65]. Additionally, ozone was found to inhibit spore germination and mediate the inactivation of cell-associated enzymes, resulting in microbial death. This blocks or inhibits the infestation of pitaya fruit by pathogenic fungi [82,83]. According to [73], rain tree leaf extracts and aqueous extracts were found to inhibit the growth of *Fusarium oxysporum*, the fungus responsible for anthracnose on pitaya. Crude extracts of ginger, turmeric rhizomes, and certain medicinal herbs were found to cause deformation, contraction, and solubilization of anthracnose mycelium at appropriate concentrations; these extracts also inhibited mycelial growth and conidial germination. In addition, the addition of 10% gum arabic (GA) to ginger or turmeric extracts at high concentrations was found to be effective in deterring anthracnose on pitaya [66,84]. Microbial growth, fruit decay, water loss, and anthracnose incidence were inhibited by a 1.0% solution of submicron chitosan dispersion (SCD) with a droplet size of 600 nm and low-molecular-weight chitosan (LMWC) [67,68].

According to [71], the growth of Streptomyces sp. in pitaya was inhibited, and the incidence of black rot was reduced by 10% phenyl ether metronidazole, 430 g/L tebuconazole, and 3% mesocarbamate [75]. In addition, it was found that UV-C irradiation effectively killed pathogenic microorganisms and reduced the incidence of black rot in pitaya. The growth of black rot spots caused by *Alternaria alternata* HP13 was inhibited by 98 mM (2.5%) sodium bicarbonate (SBC), which also prevented the fungus from infecting pitaya fruit without affecting its sensory quality [72]. Additionally, the endophytic fungus of papaya (*Penicillium rolfsii* (MK120606.1)) inhibited the mycelial growth of *Neoscytalidium dimidiatum* and enhanced the antioxidant activity of pitaya fruit [69]. Ascomycetes, which are fungi, were found to be effective in controlling the insect pest *Zophobas morio* (*Fabricius*, 1776) (*Coleoptera: Tenebrionidae*) in pitaya. They also prevented the pest from infecting the fruit and helped maintain its postharvest quality [74].

## 5. Pitaya Postharvest Preservation Technology

Presently, the preservation of pitaya after harvesting is primarily conducted through the use of physical, chemical, and biological technologies (Figure 2). These technologies serve the purpose of delaying the deterioration of postharvest quality in pitaya fruit and extending its shelf life.

### 5.1. Physical Preservation Technology

#### 5.1.1. Low-Temperature Storage

Low-temperature storage has been shown to increase the activity of protective enzymes, decrease the activity of functional enzymes, reduce the intensity of transformation and decomposition of inclusions, and maintain the quality of fruit by lowering the temperature [85]. Low-temperature storage slows down the postharvest softening of pitaya fruit scales. This effect may be attributed to the inhibition of lignin synthesis-related enzyme activities by low temperatures. Additionally, low-temperature storage increases the carotenoid content of pitaya after harvest, inhibits respiratory metabolism, prevents nutrient depletion and the occurrence of diseases, and thus helps maintain the quality of pitaya [9,86,87]. According to [88], pitaya fruit skin color, fruit hardness, total soluble solids, pH, total sugars, and total reducing sugars were optimal under 6 °C storage conditions compared to those after 14 days of storage at 6 °C, 16 °C, and 23 ± 2 °C. Maintaining a low temperature helped to preserve fruit hardness and delay nutrient consumption.

#### 5.1.2. Heat Treatment

Heat treatment is the process of killing or inhibiting pathogenic microorganisms on fruits using high temperatures ranging from 35 °C to 60 °C before storage. This helps to reduce the incidence of diseases during the storage period of fruits [89]. A 20 min treatment of pitaya fruits at 48.5 °C can reduce the occurrence of fruit flies during storage [90]. One hour of hot water treatment at 35 °C can effectively maintain the titratable acidity and hardness of fruits, resulting in better storage quality [91]. The heat treatment technique was found to improve the fruit’s reactive oxygen metabolism, enhance its antioxidant properties, and delay its aging [92].

#### 5.1.3. Controlled Atmosphere Storage

In controlled atmosphere storage, the concentration of O_2_ and CO_2_ is adjusted to alter the gas concentration in the storage environment. A gas composition with low O_2_ and high CO_2_ inhibited the respiratory metabolism of the fruit and slowed the ripening process [93]. Maintaining CO_2_ concentrations between 10.12% and 10.29% and O_2_ concentrations between 7.64% and 7.86% was found to retard increases in weight loss, rotting, malondialdehyde (MDA), and relative conductivity and to maintain total soluble solids, reducing sugar, titratable acid, and VC contents while enhancing polyphenol oxidase (PPO) activity [94].

#### 5.1.4. Irradiation

Irradiation directly affects the protein structure of microorganisms in vivo using high-energy radiation with very short wavelengths, such as gamma rays, infrared, ultraviolet, high-energy electron beams, and X-rays. This results in a change in their enzymatic activity, which in turn results in the killing of the microorganisms [95]. For temperature-dependent storage of fruits, irradiation treatments are particularly important [96]. For instance, a dose of 1.0 kJ/m^2^ UV-C was found to delay pitaya fruit scale yellowing and fruit rot [97]. Exposure to 3.2 kJ/m^2^ UV-C resulted in a significant increase in phenolic accumulation, enhanced antioxidant activity, and prevented microbial growth in a study of pitaya [98]. The application of blue light treatment, which involved exposing the fruit to 300 lx blue diode light at 450 nm for 2 h at 25 °C, significantly slowed down the increase in respiration rate, titratable acid (TA), and H_2_O_2_ content. Additionally, it reduced the content of cell wall monosaccharides, aldehydes, esters, ketones, and alkanes, which are associated with the senescence of pitaya. This treatment was found to be effective in retarding the fruit’s senescence [99].

### 5.2. Chemical Preservation Techniques

#### 5.2.1. 1-Methylcyclopropene

1-Methylcyclopropene (1-MCP) is a novel ethylene receptor blocker with several advantages, including safety, odor lessness, good stability, and ease of use. It effectively prevents both endogenous ethylene synthesis and exogenous ethylene-induced ripening and senescence [100]. The hardness, total soluble solids, respiration rate, and weight loss of pitaya were significantly reduced by the application of 600 mg/L^−1^-MCP [101]. Furthermore, the treatment with 1-MCP led to a decrease in levels of H_2_O_2_ and lipid peroxidation. Additionally, it resulted in an increase in the activities of superoxide dismutase (SOD), catalase (CAT), and ascorbate peroxidase (APX), as well as an elevation in total phenol content [102]. The application of 1-MCP delayed the rupture of the cell membrane, reduced the respiration after the harvest and preserved the quality of the pitaya fruit [103].

#### 5.2.2. Calcium Treatment

Calcium plays an important physiological role as an essential component of plant cell walls and cell membranes. Calcium treatment can be effective in stabilizing fruit color and maintaining nutritional quality [104]. Ghani [105] and Awang et al. [106] used calcium chloride to treat pitaya fruits and found that pre-harvest calcium treatment reduced the incidence of anthracnose and brown rot. The nutritional quality was unaffected, but a high concentration of calcium chloride treatment significantly increased fruit hardness while decreasing the fruit pH, TSS, and TA changes. According to [104], the application of high concentrations of calcium chloride resulted in a reduction in polygalacturonase (PG) and pectin methylesterase (PME) activities in pitaya. However, an excessive soaking time led to a decrease in hardness.

#### 5.2.3. Film Preservation

Chitosan is a commonly used substance to preserve fruits and vegetables after harvest. This treatment also reduces weight loss, delays loss of total soluble solids and titratable acidity, improves color and appearance quality, and significantly reduces postharvest disease occurrence [107]. Xing et al. (2018) found that chitosan maintained nutrient content, including soluble solids, titratable acid, and vitamin C, while somewhat reducing disease incidence. Furthermore, a specific concentration of chitosan was found to effectively prevent water loss and maintain nutrients, such as TSS, TA, and VC, during the storage of pitaya [108]. Ali A et al. [109] also discovered that double coating with chitosan significantly decreased the occurrence of pitaya rot and maintained good quality for up to 20 days of storage. Pitaya scales are unique tissue structures that result in an uneven skin surface. To effectively maintain the postharvest quality of pitaya, it is necessary to completely submerge the entire fruit surface.

#### 5.2.4. Chemical Inhibitors

Benzothiadiazole (BTH) reduced lipid peroxidation, maintained the enzyme activities of SOD, CAT, POD, and APX, upregulated the expression of HuSOD1/3/4, HuCAT2, HuAPX1/2, and HuPOD1/2/4 genes in pitaya, and increased the activities of C4H, PAL, and 4CL. This significantly delayed the senescence of pitaya [110]. Bract browning in pitaya after storage was improved by application of chloropyrifosuron (CPPU) [111]. The activity of defense-related enzymes, including phenylalanine ammonia lyase (PAL), CoA ligase (4CL), peroxidase (POD), polyphenol oxidase (PPO), chitinase (CHI), and *β*−1,3-glucanase (GLU), as well as the antifungal compounds (total phenols, flavonoids, and lignans), was increased by 0. 1 mM sodium nitroprusside solution to enhance pitaya fruit resistance to anthracnose [112]. The imidazole fungicide imidacloprid has also been used, inhibiting plasma membrane ergosterol synthesis and inducing fungal cell death [113]. Imipramine inhibited the growth of *Staphylococcus aureus* in pitaya fruit [42]. Additionally, imipramine and hot water treatment prolonged the shelf life of mango to 20 days and inhibited the development of stem-end rot and anthracnose in the absence of disease symptoms [114].

### 5.3. Biological Preservation

#### 5.3.1. Plant Hormones

Phytohormones are small organic compounds produced in minute quantities by organisms that promote or inhibit physiological processes [115]. During cold storage, salicylic acid (SA) and methyl jasmonate (MeJA) increase the antioxidant activity of pitaya fruit [116]. Furthermore, MeJA induced phenolic accumulation and prevented the reduction in ascorbic and organic acids in pitaya after trauma. The 100 μmol/L melatonin treatment increased fruit SOD, CAT, and APX activities [117,118]. Additionally, 0.1 mmol/L methyl salicylate (MeSA) regulated phenylpropanoid metabolism, promoting phenolic accumulation and activating the antioxidant system to alleviate oxidative damage caused by ROS [119]. Activation of PAL, 4CL, POD, and PPO by 10 mM β-aminobutyric acid (BABA) promoted phenolic, flavonoid, and lignin accumulation, effectively preventing pitaya fruit postharvest decay [120]. The combination of methyl jasmonate and gibberellin with chitosan was found to delay the postharvest color change in pitaya, maintain its original color, delay the reduction in vitamin C, and maintain its soluble solids content [121].

#### 5.3.2. Plant Essential Oils

Plant essential oils are becoming increasingly popular. They are safe, biodegradable, and have excellent antioxidant and bacteriostatic properties. They are suitable for postharvest preservation of fruits because of their broad-spectrum antimicrobial and bactericidal bioactivities [122]. For example, peppermint oil was found to inhibit surface mold and fungal decay in pitaya fruit by 100% over 14 days of storage. Fruit firmness, titratable acid, and total phenolic content were not affected [122]. The use of cinnamon leaf essential oil in combination with cling paper significantly delayed changes in weight loss, hardness, and peel color of pitaya during storage without significantly affecting other physiological and biochemical indices [123]. The application of natural volatile compounds resulted in a longer shelf life due to reduced microbial growth. Combining ethanol (ETOH) with methyl jasmonate (MeJA) significantly inhibited microbial growth and protected against external microbial invasion [124].

#### 5.3.3. Antagonism

Antagonistic bacteria, whose main mechanisms include inhibition of spore germination, mycelial growth, and induction of systemic resistance in plants, can be used directly or through their metabolites for postharvest preservation of fruits and vegetables [125]. *Bacillus siamensis* (*B. siamensis*) is an antagonistic strain that inhibits postharvest pathogens of tropical fruits such as mango and lychee [126]. It is also effective in preventing diseases in cabbage [127] and tomato [128] plants. Pitaya, a typical tropical fruit, showed increased disease resistance and reduced incidence and spot area when treated with *B. siamensis* N−1 compared to the control. The treatment with *B. siamensis* N−1 significantly increased the activities of phenylalanine deaminase (PAL), *β*−1,3-glucanase (GLU), chitinase (CHI), peroxidase (POD), and polyphenol oxidase (PPO). Additionally, it altered the transcriptional genes of the corresponding enzymes, which activated the postharvest disease-resistant enzyme system of pitaya fruits. This helped to avoid the excessive accumulation of ROS, thus maintaining the postharvest quality of pitaya fruit and reducing postharvest diseases [129]. The *fusaricidins* produced by *Paenibacillus polymyxa* AF01 caused irreversible damage to the membrane integrity and cellular ultrastructure of the pathogenic fungus. They directly inhibited mycelial growth, spore germination, and germ tube elongation. Additionally, they effectively inhibited *Neoscytalidium dimidiatum* growth, preventing pitaya fruit from being infected by this pathogenic fungus and maintaining the postharvest quality of pitaya [130].

## 6. Postharvest Molecular Regulation of Pitaya

Fruit ripening is a complex process involving the interaction of phytohormones, transcription factors, and epigenetic and environmental factors, involving multiple pathways such as signal transduction, energy metabolism, and substance synthesis. The molecular regulatory network controlling changes in fruit color, flavor, aroma, and texture quality indices is complex and sophisticated [131].

### 6.1. Mechanisms for Regulating Postharvest Ripening in Pitaya

#### 6.1.1. Quality of Appearance

Fruit color can visually indicate ripeness. Red-fleshed pitaya fruit contains beet pigments, mainly betacyanin and betaxanthin, with betacyanin being the primary pigment that determines the fruit’s color when it ripens [132]. The depth of the color reflects the concentration of betacyanin. The regulation of betanin biosynthesis may be influenced by the CytP450 genes HpCytP450-like1 and HpCytP450-like4 [133]. The color expression of pitaya during ripening is regulated by secondary metabolism, cytochrome P450 genes, and transcriptional changes in CYP76ADs by WRKYs [134]. The HpCYP76AD1 gene is responsible for the accumulation of the beet red pigment during pitaya ripening. It was found to be specifically up-regulated for expression and involved in the regulation of beet red pigment. Pitaya fruits exhibit different colors at different ripening stages [134]. During pitaya fruit ripening, the accumulation of betaine led to significant down-regulation of HuMYB1 and HuMYB21. Only HuMYB1, which has R2 and R3 repeats of C1, C2, C3, and C4 motifs and is localized only in the nucleus, showed transcriptional repression. This suggests that suppressing the biosynthesis of betaine inhibited fruit ripening, allowing the fruits to maintain a certain coloration [135].

Changing texture is an important indicator of ripening. During ripening or post-ripening, the texture of the fruit softens because of cell wall degradation and metabolism of cell contents. The cell wall consists mainly of polysaccharides such as pectin, cellulose, and hemicellulose, which are crosslinked with cellulose to support and protect plant structure and firmness [136]. Cell wall degrading enzymes, including polygalacturonase (PG), pectin methylesterase (PME), β-galactosidase (β-GAL), and cellulase (CEL), catalyze the breakdown of cell wall polysaccharides, leading to the softening of fruits [137]. It was discovered that during postharvest storage of pitaya, there was an increase in soluble pectin content and soluble substances, while the cellulose content decreased and the fruit’s hardness was reduced, resulting in softening. These changes in cell wall metabolism were observed [138]. Additionally, the chitosan coating inhibits the degradation of pectin polysaccharides and slows the decline in fruit firmness [139]. The firmness of fruit is influenced by the pectin polymers of chelated-soluble pectin (CSP) [140]. After harvest, flesh breakdown, water-soluble pectin (WSP), ion-soluble pectin (ISP), and covalent-soluble pectin (CSP) contents of pitaya were found to increase. Postharvest preservation maintains the postharvest quality of pitaya by delaying the depolymerization of pectin content and inhibiting the activities of polygalacturonase (PG) and cellulase (CEL) enzymes. This delays the pectin content and maintains the postharvest firmness of the fruit, maximizing its quality [141]. The inclusion of the intrinsic soluble pectin (ISP) and covalent-soluble pectin (CSP) content effectively delays the decline of non-soluble pectin (NSP) and cellulose content, thus maintaining the structural integrity of the cell wall and delaying fruit softening.

#### 6.1.2. Flavor Characteristics

Sugar plays a critical role in determining fruit flavor, and its metabolism occurs throughout the ripening process. Typically, sucrose is broken down into glucose, fructose, or UDP-glucose, which is mainly facilitated by three enzyme families: invertases (INVs), sucrose synthases (SuSys), and sucrose phosphate synthases (SPSs) [142]. Genes encoding invertases (INVs), sucrose synthases (SuSys), and sucrose phosphate synthases (SPSs) have been identified in fruits such as peach, apple, and watermelon. The expression of these enzyme family genes is closely related to sugar accumulation during fruit development and ripening and is dually regulated in sugar metabolism [143,144,145,146]. In ripe pitaya fruit, glucose is the major sugar, followed by fructose and sucrose. During the ripening of pitaya, the expression levels of sucrose hydrolase genes HpINV2 and HpSuSy1 increased significantly, which was closely related to the accumulation of glucose and fructose. Additionally, HpWRKY3 was found to be involved in sucrose accumulation by activating sucrose metabolism genes. The study showed that HpWRKY3 was positively associated with fruit sugar accumulation through the activation of sucrose metabolism genes HpINV2 and HpSUSY1 [147].

Aroma is an important intrinsic fruit quality. Aldehydes, alcohols, esters, phenols, olefins, and ketones are the primary aroma volatiles found in fruits [148,149,150,151]. The LOX pathway is the primary source of aroma volatiles, and the three key enzymes in the LOX pathway that regulate the formation of aroma substances are lipoxygenase (LOX), ethanol dehydrogenase (ADH), and lipohydroperoxide cleavage enzymes (HPL) [152]. The study found that changes in the expression of genes related to the lipoxygenase pathway, specifically FADs, LOXs, HPLs, and ADHs, may be the primary cause of the ‘mild grassy’ and ‘strong grassy’ flavors of pitaya. The analysis also revealed that the volatile components and concentrations of these genes were significant. Studies suggest that aldehydes, alcohols, esters, and olefins are most likely related to gene regulation of flavor composition, especially hexanal and 1-hexanol [153]. The aroma-related differential gene expression was mainly enriched in the fatty acids and isoleucine degradation pathways, which contributed to the specific aroma composition of pitaya.

### 6.2. Technology for Preventing and Controlling Fruit Senescence and Inducing Resistance

Ethylene is a colorless, odorless gas that is naturally produced by certain tissues and organs of plants. It acts as a plant growth regulator and promotes the ripening of fruits and vegetables [154]. The genes responsible for ethylene synthesis are the ACC synthase gene (ACS), the ACC oxidase gene (ACO), and the ACC deaminase gene (ACCD). ACS plays a critical role in ethylene synthesis, while ACO must be expressed in conjunction with ACS to be effective. The expression product of ACCD can degrade ACC, which in turn affects the ethylene content of the fruit (Figure 3) [155]. When apricot fruits at the yellow-green stage were treated with the ethylene inhibitors glycine and 1-MCP, only the expression of the ACS2 gene was significantly decreased, which was obviously affected by the two ethylene inhibitors [156]. It was hypothesized that the most critical gene in the process of ethylene synthesis was ACS2. Additionally, salicylic acid (SA) was found to effectively inhibit ethylene biosynthesis in isolated rice leaves within 2 h. Salicylic acid inhibited the conversion of ACC to ethylene but did not affect the levels of ACC and conjugated ACC [157]. Therefore, the inhibitory effect of SA was through the inhibition of ACC synthesis and the conversion of ACC to ethylene, thereby inhibiting ethylene synthesis. Controlling ethylene biosynthesis and metabolic pathways can delay fruit senescence, which is not only beneficial to postharvest quality but also improves fruit resistance to pathogenic bacteria [158]. Ethylene contributes to fruit ripening or senescence by binding to receptors called ETRs, which activate ethylene receptors and downstream signaling pathways that induce the synthesis of ripening- or senescence-related enzymes [159]. Research has shown that ethylene, as a signaling molecule, triggers an increase in phenolic, flavonoid, and related antioxidant enzyme activities at the wound site of pitaya, resulting in a reduction in reactive oxygen species (ROS) metabolism, inhibiting the upregulation of HuETR1, HuETR2, HuEIN3s, and HuERF1s and delaying fruit senescence [160]. While genes encoding MYB transcriptional repressors containing EAR motifs can directly regulate ethylene synthesis and negatively impact fruit ripening, RNA can inhibit fruit ripening and improve fruit disease resistance through RNAi-mediated SlMYB70 [161].

Melatonin regulates several aspects of pitaya quality, including weight loss, rotting rate, membrane permeability, and malondialdehyde (MDA) levels; helps maintain optimal levels of total soluble solids, ascorbic acid, and respiratory activity; and delays fruit senescence and increases the activity of enzymes associated with disease resistance (Figure 3) [135]. Furthermore, the administration of melatonin resulted in a decrease in the rate of O^2−^production, H_2_O_2_ content, and lipoxygenase activity. Additionally, it enhanced the activities of superoxide dismutase (SOD), peroxidase (POD), catalase (CAT), and ascorbate peroxidase (APX) in pitaya [118].

Another compound that has been shown to induce disease resistance in many plants is beta-aminobutyric acid (BABA). After being immersed in 10 mM BABA for 15 min and stored at ambient temperature (25 ± 2 °C, relative humidity (RH): 80–90%), the pitaya fruit showed a significant reduction in the diameter of inoculated peroxisomal lesions (Figure 3). Additionally, the treatment increased the activities of phenylalanine ammonia lyase (PAL), 4-coumarate coenzyme A ligase (4CL), peroxidase (POD), and polyphenol oxidase (PPO), promoted the accumulation of lignin, flavonoids, and phenolic compounds in the fruit, and inhibited postharvest disease infestation [120].

Gibberellin (GA) promotes cell elongation and induces amylase formation and also exhibits a pattern of “promotion followed by inhibition” of growth and respiration, according to reference [162]. Treatment with GA significantly preserved titratable acid, maintained total soluble solids, and increased vitamin C content in pitaya fruit; it also preserved the color of pitaya fruit scales, delayed cleavage of the chlorophyll-degrading enzyme, and maintained the postharvest fruit quality and nutritional status [31].

## 7. Outlook

After analyzing and summarizing previous studies, it was concluded that many studies focused on the color, physiological quality, and nutritional quality of pitaya after harvest, and various physical, chemical, and biological methods were used to control the postharvest ripening and senescence of the fruit. However, there are fewer studies on in-depth genomics (genomics, transcriptomics, proteomics, and metabolomics), and the mechanism behind extending the postharvest storage period of pitaya fruit remains unclear. The current research on postharvest preservation technology of pitaya fruit is still in its early stage and has not deeply studied the growth, development, and ripening mechanisms of the fruit.

The following aspects should be emphasized in future research on pitaya: (1) Research on pitaya cultivation technology shows that crop cultivation directly affects fruit quality. Careful planning is necessary when selecting land and fertilizers. Researchers should focus on their own research direction when selecting the base fruit to ensure the highest quality of pitaya. (2) Researching pitaya plants during the growth process of disease involves timely separation and identification of pathogenic bacterial species and screening out varieties with good resistance. Based on the study of the varieties of disease resistance gene expression, it is important to conduct an in-depth study of the gene’s ability to regulate disease resistance. This will help to achieve the purpose of prolonging the storage period of the fruit. From a comprehensive perspective, various methods such as low-temperature storage, air-conditioned storage, and coating with physical, chemical, and biological freshness techniques can be combined and explored. The use of low temperature, air conditioning, and biological methods can be considered. (3) To comprehensively store and preserve pitaya, a combination of low temperature storage, air-conditioning storage, film coating treatment, and biological preservation methods should be explored. It is important to develop postharvest storage and preservation technologies tailored to different regions and varieties of pitaya. (4) This study aims at investigating the physiological and metabolic changes that occur during postharvest storage and preservation of pitaya, with a focus on understanding the mechanisms of these changes at the cellular and molecular levels. For this purpose, we will compare pre-harvest and postharvest pitaya and samples before and after treatment by means of genomic, transcriptomic, proteomic and metabolomic analyses. The aim of this study is to improve the postharvest storage and preservation of pitaya fruit by studying the expression of related proteins or genes, key enzyme genes, and metabolic pathways involved in the ethylene synthesis process, which will help to extend the freshness and storage period of the fruit and improve its overall quality. The study will focus on pre-harvest, postharvest and pre/post treatments to identify bottlenecks in the storage and preservation industry.

## Figures and Tables

**Figure 1 foods-13-01307-f001:**
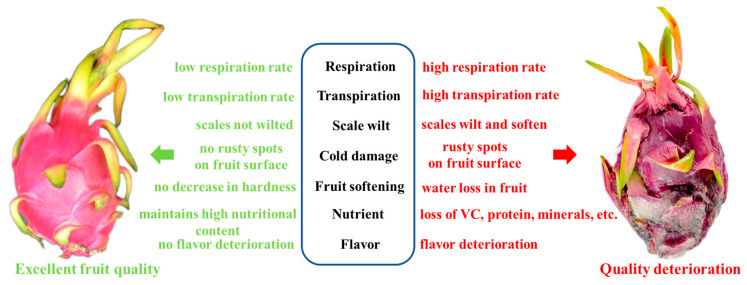
Changes in quality of pitaya.

**Figure 2 foods-13-01307-f002:**
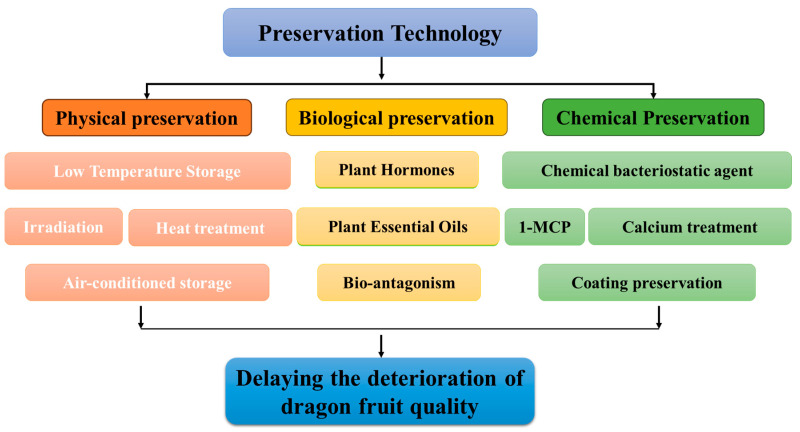
Pitaya postharvest preservation technology.

**Figure 3 foods-13-01307-f003:**
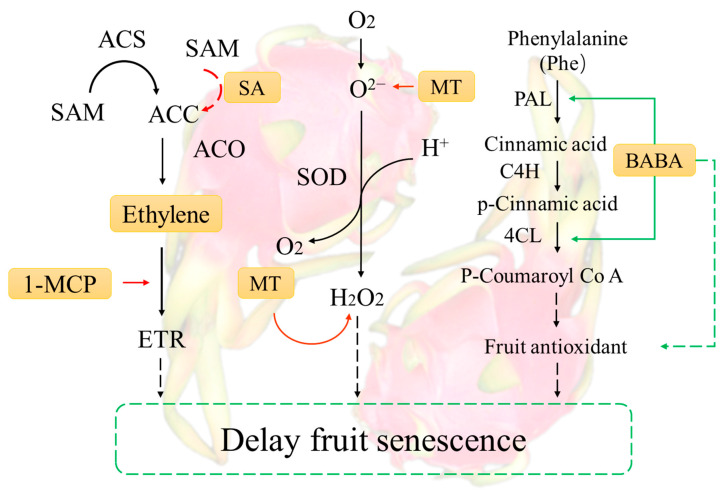
Mechanisms of phytohormone induction of postharvest ripening in pitaya (A realization indicates a direct involvement in the process of regulation, while a dotted line indicates an indirect involvement).

**Table 1 foods-13-01307-t001:** Status of pitaya.

Origin	Color Classification	Varieties	Morphological Characteristics
Israeli [1]	red-skinned fruit with red flesh with greenish scales	*Hylocereus costaricensis* or *Hylocereus polyrhizus* (Pitaya roja or red-fleshed pitaya)	A medium to large berry with red peel and green or red fleshy scales.
Colombia [15]	yellow-skinned white flesh	*Selenicereus megalanthus*	The fruit is elongated ellipsoid with green scale ends. The yellow pericarp is particularly prominent.
Antalya and Türkiye [16]	red-fleshed cultivars	Dark Star [*Hylocereus undatus*], Philippines Purple [*Hylocereus polyrhizus*], Thai Red [*Hylocereus polyrhizus*], Malaysia Red [*Hylocereus polyrhizus*], Condor [*Hylocereus guatemalensis*], American Beauty [*Hylocereus guatemalensis*])	The fruit is typically round or oval in shape, uneven in size, and has a green, scaly appearance. Red skin and either red or white flesh are common.
	white-fleshed cultivars	(‘Costa Rica White’ [*Hylocereus undatus*] and ‘Vietnamese Jaina’ [*Hylocereus undatus*]).	
Vietnam [17]	red-fleshed, oval fruit	*Selenicereus*. monacanthus	The fruit comes in varying shapes, with most being oval. It is available in three color combinations: red skin and red flesh, red skin and white flesh, and yellow skin and white flesh. The fruit varies in size, has green scales at the tips, thicker skin, and dense seeds.
	red-fleshed,	*Selenicereus* sp.	
	white-fleshed, oval fruit	*Selenicereus undatus*	
	yellow-skinned, white-fleshed, oval fruit	*Selenicereus megalanthus*	
Thai [18]	red-skinned with white flesh	Jumbo White and Vietnamese White	The fruit is round–ellipsoid in shape and variably sized, with short scales evenly distributed. The rind of the pitaya is yellow and oval with green scale tips.
	white-skinned with white flesh	Pink, Siam Red, Taiwan Red, and Ruby Red	
	yellow or golden peel white flesh	Israel Yellow	
China [19]	red peel with red pulp	Hongguan, Zhangjianghongrou, Guanhuahong, Hongshuijing	The fruit is oblong or ovoid, with red skin and either red or pink flesh, or white flesh. The scales have sharply pointed green apices, and the fruit is 10–12 cm in length with thick, waxy skin. The umbilicus is small, and the flesh appears white, red, or pink in color. The scales have green tips, and the fruit size is uneven. This type of fruit is commonly marketed as having red skin with white or red flesh.
	red-skinned with white pulp	Guanhuabai, Hongbaoshi, Yangxibairou	
	red peel and pink pulp	Guanhuahongfen	
	red peel with bicolor pulp	Shuangse	
	white-fleshed cultivars	(‘Costa Rica White’ [*Hylocereus undatus*] and ‘Vietnamese Jaina’ [*Hylocereus undatus*]).	
Bangladesh [20]	red-skinned powder flesh	*Hylocereus polyrhizus*	Common fruits found in the market include those that are oval or round in shape, have green scales, and have uneven sizes. Additionally, some have red skin with pink flesh, while others have pink skin with white flesh.
	powder-skinned white flesh	*Hylocereus undatus*	
	yellow or golden peel white flesh	Israel Yellow	
	red flesh,	*Selenicereus* sp.	
	white flesh, oval fruit	*Selenicereus undatus*	
	yellow skin, white flesh, oval fruit	*Selenicereus megalanthus*	
California [21]	red-skinned fruit with red flesh	Cebra (C, red, *Hylocereus. costaricensis*); Lisa (L, red, *Hylocereus. costaricensis*); Rosa (R, red, *Hylocereus. costaricensis*); San Ignacio (SI, red, *Hylocereus. costaricensis*);	The fruit has a thicker skin and can be oval or round with green scales on the tips. There are three variations of the fruit: one with red skin and red flesh, with seeds uniformly and densely distributed; another with red skin, pink flesh, and fewer, unevenly distributed seeds than the first variation; and a third with red skin, white flesh, and seeds densely distributed in the center of the fruit at the apex, with fewer seeds on the sides.
	red-skinned powder flesh	Physical Graffiti (PG, light pink, *Hylocereus. polyrhizus* and *Hylocereus. undatus*);	
	red-skinned, white flesh	Mexicana (M, white, *Hylocereus. undatus*)	

**Table 2 foods-13-01307-t002:** Techniques for preventing and controlling major postharvest diseases of pitaya.

Diseases	Pathogenic Fungi	Control Agents	References
Anthracnose	*Colletotrichum gloeosporioides*, Cg;*Colletotrichum runcatum*, Ct	*Bacillus* spp. CE 100	[64]
	*Colletotrichum gloeosporioides*	0.1 mM solution of sodium nitroprusside	[65]
	*Colletotrichum* pathogen	10% GA plus 15 g·L ^− 1^ ginger or turmeric extract	[66]
	*Colletotrichum* pathogen	Conventional chitosan (CC) and submicron chitosan dispersions (SCD) (1.0%, 600)	[67]
	*Colletotrichum gloeosporioides* (*Penz.*)	Crude extracts of ginger, turmeric rhizome, and ‘dukung anak’ (a medicinal herb) were used.	[68]
Ulcer disease	*Neoscytalidium dimidiatum*	Endophytic fungi of papaya (*Penicillium rolfsii* (MK120606.1))	[69]
Black rot disease	*Epicoccum sorghinum*	Bacillus subtilis (2 mL/400 mL), 2-Propanethioll (2.25 mL/400 mL), Mancozeb (2 g/400 mL), and Pyraclostrobine (1 mL/400 mL)	[70]
	*Alternaria alternata* *H8*	430 g L^−1^ tebuconazole 45% prochloraz	[71]
	*A.alternata HP13*	98 mM (2.5%) sodium bicarbonate (SBC)	[72]
	*fusarium solani*	Rainforest Leaf Extract	[73]
Other pests	Zophobas morio (*Fabricius*, 1776) (*Coleoptera: Tenebrionidae*)	Ascomycetes	[74]
	exogenous pathogens	UV-C radiation	[75]

## Data Availability

No new data were created or analyzed in this study. Data sharing is not applicable to this article.

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
