# Peer review of "Advances in the Understanding of Postharvest Physiological Changes and the Storage and Preservation of Pitaya"

_foods, 2024, doi:10.3390/foods13091307_

Round 1
Reviewer 1 Report
Comments and Suggestions for Authors
Due to the need for a comprehensive review of the increasing amount of research on pitaya fruit, I am recommending a major revision of your manuscript. The points you need to pay attention to are as follows:
1. Check that the references you cite are correct. There are far too many inappropriate or incorrect citations, making it difficult to assess the justice of your statements. (Just a very few examples: #54 refers to plums, #121 to kiwifruit, #134 to rambutan - none of them to to pitaya as reported).
2. Many statements are made about fruit suggesting that you are referring to pitaya fruit, when in fact they refer to fruit in general and give the wrong impression. For example, regarding cell wall metabolism relating to textural changes in softening fruit, no information on pitaya is presented.
3. Whole fruit preservation and fresh cut should be discussed separately.
4. The figures are either meaningless (Fig. 1) or do not relate specifically to pitaya. ( Figs. 2 and 3).
5. The detailed morphological description of fungal development of different species on PDA is superfluous.
6. Do you mean 'controlled atmosphere' when you write 'air condition'?
Comments on the Quality of English Language
Generally satisfactory with the exception of the first paragraph of the introduction, which needs to be re-written.
Author Response
Dear reviewer,
Thank you so much for taking the time to provide me with your valuable suggestions on my paper. I really appreciate your feedback and have incorporated your changes into the document. To make it easier for you to identify the changes, I used a uniform gray fill for the font background color. Let me know if you have any further comments or suggestions. Thank you again for your help! Please find the revised version attached below.
- Check that the references you cite are correct. There are far too many inappropriate or incorrect citations, making it difficult to assess the justice of your statements. (Just a very few examples: #54 refers to plums, #121 to kiwifruit, #134 to rambutan - none of them to to pitaya as reported)
Reply 1 : Dear reviewer, I have made changes as per your suggestions.
- Many statements are made about fruit suggesting that you are referring to pitaya fruit, when in fact they refer to fruit in general and give the wrong impression. For example, regarding cell wall metabolism relating to textural changes in softening fruit, no information on pitaya is presented.
Reply 2 : Dear reviewer, I have made changes as per your suggestions. I have made the changes you suggested as follows. Thank you again.
- Whole fruit preservation and fresh cut should be discussed separately.
Reply 3 : Dear reviewer, Thank you. Based on the valuable suggestions you have made I have already made changes to the part of the article on freshly cut dragon fruits after picking already
4.The figures are either meaningless (Fig. 1) or do not relate specifically to pitaya. ( Figs. 2 and 3).
Reply 4 : Dear reviewer, Thank you. The purpose of Figures 1 and 2 is to visualize the postharvest quality changes of dragon fruit and to illustrate the postharvest preservation techniques. Figure 3 summarizes the effects of antioxidants and ROS-related pathways on the postharvest quality of dragon fruit based on the current postharvest study.
5.The detailed morphological description of fungal development of different species on PDA is superfluous.
Reply 5 : Dear reviewer, I have deleted the expression pathogenic bacteria and revised it as follows:
- Do you mean 'controlled atmosphere' when you write 'air condition'?
Reply 6 : Dear reviewer I have revised the inappropriate expressions and thank the reviewer for his valuable suggestions!
Reply 7:
The following changes have been made to the introduction:

Reviewer 2 Report
Comments and Suggestions for Authors
This review on “Advances in the understanding of postharvest physiological changes and the storage and preservation of dragon fruit” systematically describes the postharvest quality changes of pitaya as well as the disease and further analyzed and summarized for the postharvest preservation technology of pitaya fruit, and finally elaborated the molecular basis of postharvest ripening and senescence as well as resistance induction of pitaya, which was relatively systematic. However, there are still grammatical and formatting problems in this review, and we hope that the authors will revise it carefully.
1. In line 3, change dragon fruit to pitaya, and rewrite the entire text.
2. Abstract is usually written in the past tense and is checked to ensure that the tenses used match.
3. Table 1: Brief description of pitaya morphological characteristics.
4. The abbreviation Vegetable oil (VOC) in line number 103 should be given in full the first time it appears.
5. References [39,40,41,42,43] in line number 148 and [2, 5-8] in line number 167 should appear in one format throughout the text.
6. Figure 1: Uniformity of quality variations in the pictures, checking whether the effect of certain quality variations on the overall quality of the fruit is consistent.
7. Table 2: Use the full name of the fungus in the text as much as possible, and check the whole text.
8. Please standardize the writing of H2O2 in line number 327.
9. The final mechanism should be optimized, and the base image should be the original image.
10. The abbreviations of some antioxidant enzymes appear in line number 349, and the first time they appear, the full name should be written before the abbreviation.
11. as a result of represents the result, emphasizing the reason why the result of something is. is often used in result adverbial clauses. " because of " means cause, emphasizing what causes an event or result to occur. Check the sentence with line number 446(Changing texture is an important indicator of ripening. During ripening or post-ripening, the texture of the fruit softens as a result of cell wall degradation and metabolism of cell contents.)
12. Please check the spelling of the word endoglycosyltransferases in line number 454, and check the spelling of the terminology throughout the text to ensure that it is written correctly. Same for lipohydroperoxidative in line number 485.
13. Go through 4.2 Techniques to Prevent and Control Disease to ensure that the table 2 prevention and control techniques are covered in the text.
14. Check whether the references are cited repeatedly.
15. For the format of references, please refer to the requirements of this journal's submission guidelines for modification.
Author Response
Dear reviewer,
Thank you so much for taking the time to provide me with your valuable suggestions on my paper. I really appreciate your feedback and have incorporated your changes into the document. To make it easier for you to identify the changes, I used a uniform yellow fill for the font background color. Let me know if you have any further comments or suggestions. Thank you again for your help! Please find the revised version attached below.
- In line 3, change dragon fruit to pitaya, and rewrite the entire text.
Reply 1 : Dear reviewer, I have made changes as per your suggestions.
- Abstract is usually written in the past tense and is checked to ensure that the tenses used match.
Reply 2 : Dear reviewer,I have checked the abstract section and the abstract section has been modified according to your suggestions
- Table 1: Brief description of pitaya morphological characteristics.
Reply 3 :
Origin |
Color classification |
Varieties |
Morphological characteristics |
Israeli[1] |
red-skinned fruit with red flesh with greenish scales |
Hylocereus costaricensis or Hylocereus polyrhizus (Pitaya roja or red-fleshed pitaya) |
A medium to large berry with red peel and green or red fleshy scales. |
Colombia[15] |
yellow-skinned white flesh |
Selenicereus megalanthus |
The fruit is elongated ellipsoid with green scale ends. The yellow pericarp is particularly prominent. |
Antalya and Türkiye[16]
|
red-fleshed cultivars |
Dark Star [Hylocereus undatus], Philippines Purple [Hylocereus polyrhizus], Thai Red [Hylocereus polyrhizus], Malaysia Red [Hylocereus polyrhizus], Condor [Hylocereus guatemalensis], American Beauty [Hylocereus guatemalensis]) |
The fruit is typically round or oval in shape, uneven in size, and has a green, scaly appearance. Red skin and either red or white flesh are common. |
white-fleshed cultivars |
(‘Costa Rica White’ [Hylocereus undatus] and ‘Vietnamese Jaina’ [Hylocereus undatus])。 |
||
Vietnam [17] |
red-fleshed, oval fruit |
Selenicereus. monacanthus |
The fruit comes in varying shapes, with most being oval. It is available in three color combinations: red skin and red flesh, red skin and white flesh, and yellow skin and white flesh. The fruit varies in size, has green scales at the tips, thicker skin, and dense seeds. |
red-fleshed, |
Selenicereus sp. |
||
white-fleshed, oval fruit |
Selenicereus. undatus |
||
yellow-skinned, white-fleshed, oval fruit |
Selenicereus. megalanthus |
||
Thai [18] |
red-skinned with white flesh
|
Jumbo White and Vietnamese White. |
The fruit is round-ellipsoid in shape and variable sized, with short scales evenly distributed. The rind of the pitaya is yellow and oval with green scale tips.
|
white-skinned with white flesh |
Pink, Siam Red, Taiwan Red, and Ruby Red. |
||
yellow or golden peel white flesh |
Israel Yellow |
||
China [19] |
red-peel with red-pulp |
Hongguan, Zhangjianghongrou, Guanhuahong, Hongshuijing |
The fruit is oblong or ovoid, with red skin and either red or pink flesh, or white flesh. The scales have sharply pointed green apices, and the fruit is 10-12 cm in length with thick, waxy skin. The umbilicus is small, and the flesh appears white, red, or pink in color. The scales have green tips, and the fruit size is uneven. This type of fruit is commonly marketed as having red skin with white or red flesh. |
red-skinned with white-pulp |
Guanhuabai, Hongbaoshi, Yangxibairou |
||
red-peel and pink-pulp |
Guanhuahongfen |
||
red-peel with bicolor-pulp |
Shuangse |
||
white-fleshed cultivars |
(‘Costa Rica White’ [Hylocereus undatus] and ‘Vietnamese Jaina’ [Hylocereus undatus])。 |
||
Bangladesh [20]
|
red-skinned powder flesh |
Hylocereus polyrhizus |
Common fruits found in the market include those that are oval or round in shape, have green scales, and have uneven sizes. Additionally, some have red skin with pink flesh, while others have pink skin with white flesh. |
powder-skinned white flesh |
Hylocereus undatus |
||
yellow or golden peel white flesh |
Israel Yellow |
||
red flesh, |
Selenicereus sp. |
||
white flesh, oval fruit |
Selenicereus. undatus |
||
yellow skin, white flesh, oval fruit |
Selenicereus. megalanthus |
||
California [61] |
red-skinned fruit with red flesh |
Cebra (C, red, Hylocereus. costaricensis); Lisa (L, red, Hylocereus. costaricensis); Rosa (R, red, Hylocereus. costaricensis); San Ignacio (SI, red, Hylocereus. costaricensis); |
The fruit has a thicker skin and can be oval or round with green scales on the tips. There are three variations of the fruit: one with red skin and red flesh, with seeds uniformly and densely distributed; another with red skin, pink flesh, and fewer, unevenly distributed seeds than the first variation; and a third with red skin, white flesh, and seeds densely distributed in the center of the fruit at the apex, with fewer seeds on the sides. |
red-skinned powder flesh |
Physical Graffiti (PG, light pink, Hylocereus. polyrhizus and Hylocereus. undatus); |
||
red-skinned, white flesh, |
Mexicana (M, white, Hylocereus. undatus). |
- The abbreviation Vegetable oil (VOC) in line number 103 should be given in full the first time it appears.
Reply 4:Dear reviewer, I have made changes as per your suggestions.
- References [39,40,41,42,43] in line number 148 and [2, 5-8] in line number 167 should appear in one format throughout the text.
Reply 5:Dear reviewer, I have changed the format.
- Figure 1: Uniformity of quality variations in the pictures, checking whether the effect of certain quality variations on the overall quality of the fruit is consistent.
Reply 6:Dear reviewer, Hello! I have verified that Figure 1 is correctly depicted
- Table 2: Use the full name of the fungus in the text as much as possible, and check the whole text
Reply 7:Dear reviewer, I have checked and made changes as per your suggestions.
- Please standardize the writing of H2O2 in line number 327.
Reply 8:Thank you. I have made changes.
- The final mechanism should be optimized, and the base image should be the original image
Reply 9: Dear reviewer I have revised and thank the reviewer for his valuable suggestions!
- The abbreviations of some antioxidant enzymes appear in line number 349, and the first time they appear, the full name should be written before the abbreviation.
Reply 10: Dear reviewer, I have checked and made changes as per your suggestions.
- as a result of represents the result, emphasizing the reason why the result of something is. is often used in result adverbial clauses. " because of " means cause, emphasizing what causes an event or result to occur. Check the sentence with line number 446(Changing texture is an important indicator of ripening. During ripening or post-ripening, the texture of the fruit softens as a result of cell wall degradation and metabolism of cell contents.)
Reply 11: Dear reviewer I have made changes in the result and thank the reviewer for his valuable suggestions!
- Please check the spelling of the word endoglycosyltransferases in line number 454, and check the spelling of the terminology throughout the text to ensure that it is written correctly. Same for lipohydroperoxidative in line number 485.
Reply 12: Dear reviewer, I have checked and made changes as per your suggestions.
- Go through 4.2 Techniques to Prevent and Control Disease to ensure that the table 2 prevention and control techniques are covered in the text.
Reply 13 :Dear reviewer, I have verified that table 2 is correctly depicted
- Check whether the references are cited repeatedly.
Reply 14 : Dear reviewer, I have verified that table 2 is correctly depicted
- For the format of references, please refer to the requirements of this journal's submission guidelines for modification.
Reply 15 : Dear reviewer, I have verified the references.

Reviewer 3 Report
Comments and Suggestions for Authors
The review on post harvest preservation and handling of dragon fruit is well written. The manuscript covers physical, chemical and biological methods for both handling and preservation of dragon fruits. Addition of some more details on following points will benefit the impact of the review.
1. Are there any hurdle technologies that are being considered by the researchers to store and preserve dragon fruits?
2. Include the review of research works on combination of the physical, chemical and biological methods for storage and preservation of dragon fruits.
Author Response
Dear reviewer,
Thank you very much for your valuable comments on my dissertation in your busy schedule. I am very much appreciative of your feedback. I have listed and responded below to the responses related to your valuable suggestions. Please let me know if you have any other comments or suggestions. Once again, thank you for your help!
- Are there any hurdle technologies that are being considered by the researchers to store and preserve dragon fruits?
Reply1:
Dear Reviewer, Hello! The technology for preserving the postharvest quality of dragon fruit is currently limited to physiological changes, with little research on the underlying mechanisms. Specifically, there is a lack of studies on the histological changes related to sugar metabolism, flavonoid metabolism, polyamine metabolism, and antioxidant systems. The challenges facing dragon fruit postharvest preservation technology are primarily twofold: (1) Theunique phenological structure of dragon fruit significantly reduces scale protection during storage and preservation; and (2) There isa lack of research on the postharvest cell wall metabolism of dragon fruit and its effect on fruit softening through histologyï¼›(3) Further investigation is needed to understand the mechanism behind the pre-harvest and post-harvest flavor changes in dragon fruit.
- Include the review of research works on combination of the physical, chemical and biological methods for storage and preservation of dragon fruits.
Reply 2:
Dear Reviewer, Hello! Once again, thank you for your valuable suggestions. This paper reviews the postharvest physical, chemical, and biological technologies of dragon fruit separately. However, due to limitations in current research, studies involving the combination of these technologies were not reviewed. The review also discusses future developments and explorations in dragon fruit postharvest preservation technologies.

Round 2
Reviewer 3 Report
Comments and Suggestions for Authors
The authors have significantly improved the manuscript and have addressed all the comments raised by the reviewer. Thus the manuscript may be considered for publication.
Author Response
Dear reviewer,
I would like to express my gratitude for your invaluable feedback on my thesis. Your insights have been instrumental in guiding me towards a more robust and compelling argument. I am deeply appreciative of your support and encouragement.
Once again, I would like to thank you sincerely for dedicating your time to reviewing my thesis.